# A TriNetX Registry Analysis of the Need for Second Procedures following Index Anterior and Posterior Urethroplasty

**DOI:** 10.3390/jcm12052055

**Published:** 2023-03-05

**Authors:** Zachary J. Prebay, Adam M. Ostrovsky, Matthew Buck, Paul H. Chung

**Affiliations:** Department of Urology, Sidney Kimmel Medical College, Thomas Jefferson University, Philadelphia, PA 19107, USA

**Keywords:** urethral stricture, reintervention, urethroplasty, endoscopic treatment

## Abstract

Background: We queried a global database to understand re-intervention rates following urethroplasty with the goal of evaluating whether they align with previously published data. Methods: Using the TriNetX database and Common Procedural Terminology (CPT) and International Classification of Diseases-10 (ICD) codes, we identified adult male patients with urethral stricture disease (ICD N35) who underwent one-stage anterior (CPT 53410) or posterior urethroplasty (CPT 53415), with or without (substitution urethroplasty) a tissue flap (CPT 15740) or buccal graft (CPT 15240 or 15241). We set urethroplasty as the index event and used descriptive statistics to report the incidence of secondary procedures (using CPT codes) within 10 years after the index event. Results: There were 6606 patients who underwent urethroplasty within the last 20 years, with 14.3% of patients undergoing a second procedure after index event. Upon subgroup analysis, reintervention rates were 14.5% for anterior urethroplasty vs. 12.4% of patients with an anterior substitution urethroplasty (RR 1.7, *p* = 0.09) and 13.3% for posterior urethroplasty vs. 8.2% for patients with a posterior substitution urethroplasty (RR 1.6, *p* < 0.01). Conclusions: Most patients will not need any form of re-intervention following urethroplasty. These data align with previously described recurrence rates, which may help urologists counsel patients considering urethroplasty.

## 1. Introduction

Urethral stricture disease (USD) can result from a variety of causes, including infectious, inflammatory, traumatic, idiopathic, and, unfortunately, iatrogenic factors [1,2]. USD is managed primarily through endoscopic intervention, either direct visualization internal urethrotomy (DVIU) or dilation, or through surgical intervention [3]. Surgical repair involves urethroplasty, which can be performed with multiple stages and techniques, such as fasciocutaneous flaps or mucosal grafts, in differing configurations [4,5,6]. Following urethroplasty, postoperative complications may include erectile and/or ejaculatory dysfunction [7], stress incontinence [8], or recurrent stricture. A common definition of success following urethroplasty is no further need for intervention or instrumentation [9]. The success rate varies based on factors, such as stricture location, technique performed, and time of follow-up, but the overall patency rate tends to be high, with estimates around 75% for all types of single-stage graft urethroplasties for penile urethral strictures [4], above 90% for bulbar strictures [1,10], and around 85% for posterior urethral stenosis [11,12].

However, the data on outcomes following urethroplasty have notable limitations, as most of the data regarding outcomes following urethroplasty come from single-surgeon or single-institutional studies. Further, these studies tend to be from highly regarded surgeons or centers of excellence in reconstructive urology. Therefore, urologists may rightly be cautious quoting these data to patients as the generalizability to “real-world” centers and practice may justly be questioned. In the official American Urological Association guidelines on male stricture disease, the authors recommend further multi-institutional studies to evaluate the outcomes of treatment for USD, and additionally for studies that assess outcomes following different urethroplasty techniques, such as with or without a flap or graft [2].

In this context, we sought to utilize the large, multi-institutional TriNetX database to evaluate the need for repeat intervention following urethroplasty. USD is a broad and complex disease; thus, it can be challenging to give a specific prediction of any given patient’s postoperative outcome. However, with the vast wealth of data within TriNetX, we hope to provide general estimates of the common patient’s success following urethroplasty. We hypothesize the reintervention rate within TriNetX will closely approximate the available rates quoted in the literature, which may help providers when counseling patients to provide the most accurate estimation of their risk for repeat intervention.

## 2. Materials and Methods

We accessed the TriNetX Research Network, a collaborative research enterprise that contains real-time data from the electronic health records of over 100 million patients located in 71 healthcare organizations across the globe at time of analysis. TriNetX analyzes patient data up to 20 years prior to the date of analysis (2002–2022), therefore, excluding those undergoing the index event over 20 years ago, and includes data on demographics, medical diagnoses, procedures, lab values, and medications. Given the de-identified nature of this dataset, our study was deemed exempt from Institutional Review Board approval. We identified our cohort using a combination of Current Procedural Terminology (CPT) and International Classification of Diseases-10 (ICD) codes.

Our initial cohort identified all adult (greater than or equal to 18 years old) male patients with USD (ICD N35) who underwent one-stage anterior (CPT 53410) or posterior urethroplasty (CPT 53415), with or without a tissue flap (CPT 15740) or buccal graft (CPT 15240 or 15241) (“substitution urethroplasty”). We choose to only evaluate initial one-stage urethroplasty for our index event as these are most common, and the added morbidity and complexity of staged repairs would make our results less generalizable and deserve dedicated study. Anterior and posterior USD was determined by the CPT codes within TriNetX and anatomically divided between the bulbar and membranous urethra.

The index event is urethroplasty and we use descriptive statistics to report the incidence of secondary procedures within 10 years after index event for our overall population and each sub-category. Our primary outcome is overall incidence of secondary procedures with secondary outcomes of endoscopic reintervention and surgical reintervention. Endoscopic reintervention included cystoscopy and dilation (CPT 52281), direct visualization internal urethrotomy (CPT 52275 or 52276), and dilation with sounds (CPT 53600, 53601, 53605); surgical reintervention included CPT 53400, 53410, and 53415. Patients who underwent a second urethroplasty had this surgery reset for index event.

Additionally, we compared between anterior and anterior substitution urethroplasty and posterior and posterior substitution urethroplasty for risk of reintervention, with statistics reported in terms of Risk Ratio (RR). Statistical significance was set at a *p*-value of 0.05.

We also performed Kaplan–Meier analysis for time to overall reintervention for the overall population and each comparison sub-analysis. This is reported in terms of projected 10-year survival probability and comparisons were made using Hazard Ratios (HRs) with 95% confidence intervals. Additional graphics were generated in R (R Foundation for Statistical Computing, Vienna, Austria) with the aid of the tidyverse package [13].

## 3. Results

Analyses were run on 4 January 2023. There were 6606 patients who underwent urethroplasty within the last 20 years, with 14.3% of patients undergoing a second procedure after the index event. Patients who underwent urethroplasty were, on average (mean), 49.1 years old at index event, with a standard deviation of 16.9 years. Most patients were White (69%), followed by Unknown (15%) and Black (14%). At 10 years, the Kaplan–Meier analysis showed that projected survival probability (not needing a second intervention) was 70.3%.

There were 3401 patients who underwent anterior urethroplasty and 1079 patients who underwent anterior substitution urethroplasty. Patients were similar ages at index event (49.4 years for anterior urethroplasty and 49.3 years for anterior substitution urethroplasty). Most patients were White (66% and 76% for anterior and anterior substitution). Patients with anterior urethroplasty had a reintervention rate of 14.5% and those with anterior substitution urethroplasty had a reintervention rate of 12.4% (RR 1.7, *p* = 0.09) (Table 1). For the anterior and anterior substitution urethroplasty approaches, respectively, endoscopic reintervention rates were 10.2% and 9.0% (RR 1,1, *p* = 0.26) and surgical reintervention rates were 6.9% and 6.2% (RR 1.1, *p* = 0.41), respectively. Kaplan–Meier analysis showed projected 10-year survival probability of 72.2% and 72.5% for anterior and anterior substitution urethroplasty (HR 1.1, 95%CI 0.91–1.3) (Figure 1).

Similarly, there were 2148 patients who underwent posterior urethroplasty and 430 patients who underwent posterior substitution urethroplasty. Patients with posterior urethroplasty were also of similar ages, regardless of approach (49.1 years and 49.5 years for substitution urethroplasty). Again, most patients were White (72% and 82% for substitution). Overall reintervention rates were 13.3% and 8.2%, respectively (RR 1.6, *p* < 0.01). Endoscopic reintervention rates were 10.5% and 6.5% (RR 1.6, *p* = 0.01) and surgical reintervention rates were 4.7% and 3.7% (RR 1.3, *p* = 0.39). Upon Kaplan–Meier analysis, projected survival probability at 10 years was 74.3% and 79.6% for posterior and posterior substitution urethroplasty (HR 1.6, 95% CI 1.1–2.3) (Figure 2).

## 4. Discussion

Stricture recurrence can occur at any point in the postoperative period; thus, there is no clear consensus on postoperative follow-up [2]. A combination of flow analysis, post-void residual, cystoscopy, ultrasound, and urethrogram can be used to determine stricture recurrence [1,2]. There is also debate on how to best define success following urethroplasty. The European guidelines distinguish between the traditional “academic” definition (lack of reintervention) and the alternative “anatomic” definition (normal urethral lumen). Both of these definitions are useful adjuncts to the true measure of success following urethroplasty, which should be patient-based outcomes, such as satisfaction and quality of life [14,15]. Our work relied upon the objective definition of reintervention to determine success. This is necessary because TriNetX, like most administrative datasets, does not contain data on the subjective patient experience. Despite missing this key component, our hope is that by more accurately describing reintervention rates following urethroplasty, we can provide more accurate patient counseling, which, in turn, may result in better patient expectations and more satisfaction with postoperative outcomes.

In this study, we describe the rates of overall reintervention as well as endoscopic and surgical reintervention following urethroplasty for male USD. In the entire cohort, 14.3% of patients underwent a reintervention with the projected 10-year reintervention-free survival at 70.6%. This overall reintervention largely aligns with previous literature. Breyer et al. reported on long-term success following urethroplasty with 21% of patients in their cohort having stricture recurrence at a median follow-up of 5.8 years [16]. However, although they included over 400 patients, this was a single-surgeon series at a recognized center of excellence for urologic reconstruction, which, by itself, may give pause to other urologists regarding the generalizability of their success. It is important to have contemporary data regarding urethroplasty success, especially given increased emphasis by societal guidelines to recommend urethroplasty if endoscopic attempts fail [17].

Interestingly, patients who underwent substitution urethroplasty including a flap or graft had statistically significant lower overall reintervention rates compared to those who did not have a flap or graft for both anterior and posterior approaches. The data also suggest that patients with an anterior stricture have numerically higher reintervention rates than patients with a posterior stricture when comparing between patients with and without flaps or grafts. Without the necessary patient or operative details that TriNetX does not provide, caution must be taken when making this interpretation, as differences in patient disease may wholly or partially account for this finding. It is also possible that patients with posterior strictures were more likely to undergo treatment with intermittent catheterization, which would not be captured in this analysis.

One distinction in our study is the relatively higher rate of repeat urethroplasty as opposed to endoscopic reintervention compared to other series. Kahokehr et al. looked at their institutional analysis of almost 400 patients with bulbar urethroplasty performed by two surgeons. Their recurrence rate was low, only 25 patients, and almost all of these (92%) were initially managed endoscopically [18]. For our analysis, we combined all grafts and flaps in our study. This was based on the inability to distinguish exact surgical techniques and details in TriNetX. This is supported by the literature, which has noted fairly comparable success rates (typically >80%), irrespective of which flap or graft is used [19,20,21,22], similar to our findings.

While our results are broadly generalizable, we note limitations that may limit a provider’s ability to counsel specific patients. To begin, this is a retrospective and descriptive study with the data integrity outside the control of the investigators. This includes that some patients may have been lost to follow-up for various reasons, including reestablishing care with a healthcare organization outside of the TriNetX network. Within TriNetX, there are no specific patient and operative descriptors, and our analyses were limited to generic ICD and CPT codes. Further, without access to the raw data, we were unable to perform additional or separate analyses from those within TriNetX. The limitations of our study also restrict our ability to use the results to guide clinical practice. For example, we lack data on the size of stricture, both primary and recurrent, the severity of the stricture, as well as the size of graft or flap used and the indication for flap or graft. These factors are critical for preoperative planning as a surgeon decides which approach is correct for each patient. We, thus, caution interpretation of our comparisons between urethroplasty and substitution urethroplasty as our study lacks the stricture characteristics that are at least partially responsible for the outcomes presented. Additionally, we defined our definition of treatment failure as need for reintervention, and because there exist different ways to define treatment success or failure after urethroplasty, this may not be the best way to compare outcomes following urethroplasty. However, we believe the need for reintervention is a clinically significant outcome that is easy for patients to understand. As TriNetX does not publish the list of health-care organizations that are included in the network, we are unable to assess how many fellowship-trained providers may be included in this analysis.

Despite these limitations, our study has notable strengths, primarily the size and multi-institutional nature of our patient population. Having multi-institutional or registry-based outcomes as opposed to single- or few-institutional studies has been lacking in male USD literature. USD is very complex, with a broad range of patient and stricture characteristics. It is important for surgeons to be experienced with various techniques and principles to provide the best care for their patients. We show that across a wider patient population, most patients undergoing urethroplasty experience successful outcomes. These data, therefore, accomplish our goal of establishing general success rates, which can be used to counsel patients who are considering single-stage urethroplasty.

## 5. Conclusions

In conclusion, we describe a contemporary and multi-institutional registry-based analysis of male USD and rates of reintervention following urethroplasty. We further divided our analysis into those undergoing an anterior or posterior approach and with or without a flap or graft. Overall, most patients did not require reintervention. The limitations of this study prevent specific inferences from being applied to individual patients, but on a broader scale, the data herein can help providers put expected patient outcomes into context.

## Figures and Tables

**Figure 1 jcm-12-02055-f001:**
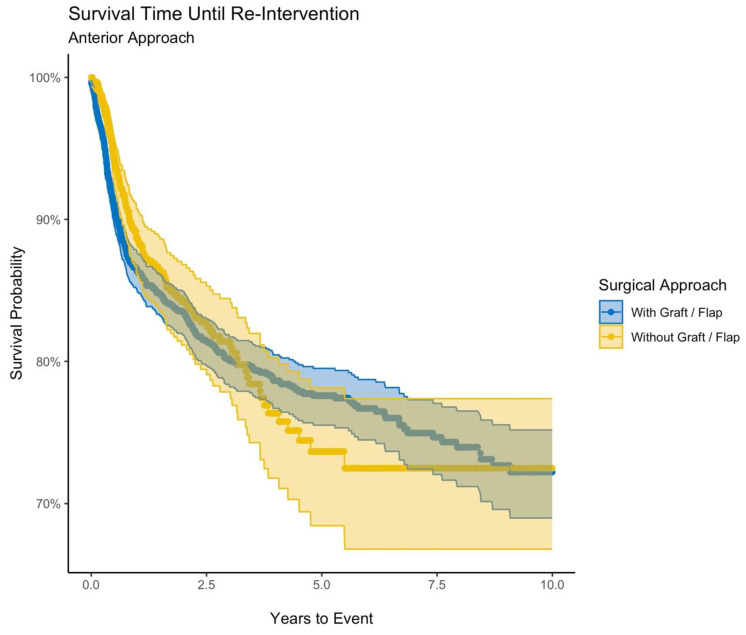
Survival time until reintervention (anterior approach).

**Figure 2 jcm-12-02055-f002:**
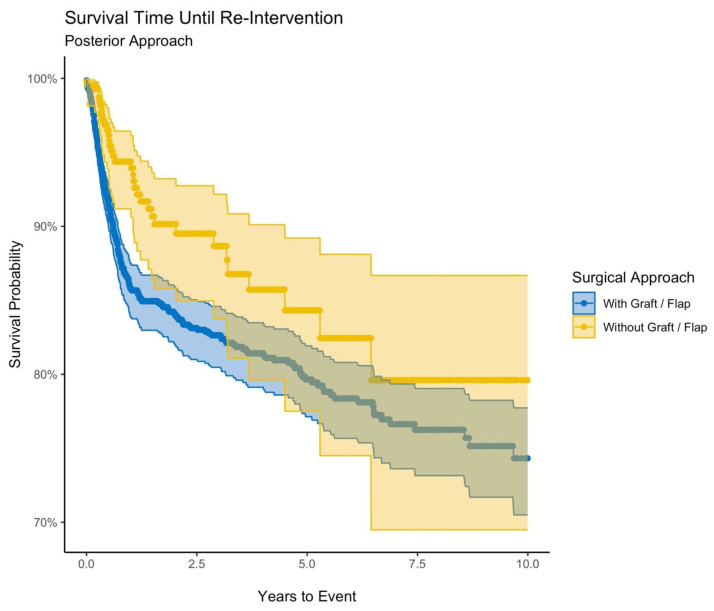
Survival time until reintervention (posterior approach).

**Table 1 jcm-12-02055-t001:** Rates of secondary procedures performed after index urethroplasty.

	Overall Re-Intervention	Surgical Re-Intervention	Endoscopic Re-Intervention
Anterior Urethroplasty without tissue flap or buccal graft n = 3243	14.5%	6.9%	10.2%
Anterior Urethroplasty with tissue flap or buccal graft n = 877	12.4%	6.2%	9.0%
Posterior Urethroplasty without tissue flap or buccal graft n = 2073	13.3%	4.7%	10.5%
Posterior Urethroplasty with tissue flap or buccal graft n = 414	8.2%	3.7%	6.5%

## Data Availability

Data proprietary to TriNetX.

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
