# Peer review of "A TriNetX Registry Analysis of the Need for Second Procedures following Index Anterior and Posterior Urethroplasty"

_jcm, 2023, doi:10.3390/jcm12052055_

Round 1

Reviewer 1 Report

Congratulations to the authors for this fine article. 

The topic is highly interesting and is well presented. 

There are some points, the authors should highlight in their manuscript. 

-       The differentiation between anterior and posterior USD is not clear. 

-       Although the following points cannot be evaluated by using TriNetX Registry Analysis, it is important to highlight these issues in the discussion: 

-       One of the most critical points in performing urethroplasty as a surgeon is the knowledge about the individual indication for the correct surgical approach in each patient.

-       Indication which kind of USD-surgery to perform depends on stricture-length, -severity, -etiology, spongiofibrosis, etc. To choose the correct technique defines the success rate. It is very likely that this could be one reason, why the non-substitutional urethroplasties have a lower success rate as well as the anterior stricture-surgeries.

-     The issue, that the intervention indication is not clearly evaluable has to be more highlighted in the discussion, although it is already mentioned by the authors. 

-       in my opinion the conclusion is not appropriate. USD surgery is very complex and a broad knowledge about it is a prerequisite for the success of the procedure. To provide best postoperative outcome, it is obligatory to know different surgical techniques and principles and to have the ablility to switch between them depending on the intraoperative course. The knowledge of this complex disease, its anatomical features, and the therapies - appropriate to the respective situations - are decisive.

Author Response

Congratulations to the authors for this fine article. 

The topic is highly interesting and is well presented. 

We thank the Reviewer for their kind words.

There are some points, the authors should highlight in their manuscript. 

-       The differentiation between anterior and posterior USD is not clear. 

We have attempted to clarify this distinction in the methods section.

Anterior and Posterior USD was determined by the CPT codes within TriNetX and anatomically is divided between the bulbar and membranous urethra, respectively.”

-       Although the following points cannot be evaluated by using TriNetX Registry Analysis, it is important to highlight these issues in the discussion: 

-       One of the most critical points in performing urethroplasty as a surgeon is the knowledge about the individual indication for the correct surgical approach in each patient.

-       Indication which kind of USD-surgery to perform depends on stricture-length, -severity, -etiology, spongiofibrosis, etc. To choose the correct technique defines the success rate. It is very likely that this could be one reason, why the non-substitutional urethroplasties have a lower success rate as well as the anterior stricture-surgeries.

-     The issue, that the intervention indication is not clearly evaluable has to be more highlighted in the discussion, although it is already mentioned by the authors. 

We appreciate the Reviewer's thoughtful comments. These important considerations do certainly limit interpretation of these findings and how to apply them to clinical practice. We are of the opinion that our results are still of descriptive benefit to surgeons and patients, but certainly must be interpreted with caution if choosing to let our results guide surgical planning. We have more strongly emphasized this point in our discussion, particularly the limitations paragraph.

Also, without access to the raw data we were unable to perform additional or separate analyses from those within TriNetX. The limitations of our study also restrict our ability to use the results to guide clinical practice. For example, we lack data on the size of stricture, both primary and recurrent, the severity of the stricture, as well as the size of graft or flap used and the indication for flap or graft. These factors are critical for preoperative planning as a surgeon decides which approach is correct for each patient. We thus caution interpretation of our comparisons between urethroplasty and substitution urethroplasty as our study lacks the stricture characteristics that are at least partially responsible for the outcomes presented.”

-     in my opinion the conclusion is not appropriate. USD surgery is very complex and a broad knowledge about it is a prerequisite for the success of the procedure. To provide best postoperative outcome, it is obligatory to know different surgical techniques and principles and to have the ablility to switch between them depending on the intraoperative course. The knowledge of this complex disease, its anatomical features, and the therapies - appropriate to the respective situations - are decisive.

The Reviewer again makes strong points. We have amended our discussion to emphasize the points raised above.

“USD is very complex with a broad range of patient and stricture characteristics. It is important for surgeons to be experienced with various techniques and principles to provide the best care for their patients. We show that across a wider patient population, most patients undergoing urethroplasty experience successful outcomes. This data therefore accomplishes our goal of establishing general success rates which can be used to counsel patients who are considering single stage urethroplasty.”

We also have established in our introduction that our goal in this manuscript is to provide a general idea of success following urethroplasty with the understanding that given the complexity of USD it wouldn’t be fair to use our data to predict success with all combinations of surgical and stricture attributes.

USD is a broad and complex disease thus it can be challenging to give a specific prediction of any given patient’s postoperative outcome. However, with the vast wealth of data within TriNetX, we hope to provide general estimates of the common patient’s success following urethroplasty.”

Reviewer 2 Report

A simple posthoc Analysis of Need for Second Procedures following Index Anterior and Posterior Urethroplasty. The finding is meaningful but not strong enough.

The reality of this data is needed to be verified. Authors should upload the data as the supplement.

Demographics and clinical characteristics of the study participants according to Anterior and Posterior Urethroplasty are missed.

CPTs are not needed to show in the table. x2 test should be perfomed in table 1.

KM figures listed a very wide 95% CI. Detailed numbers of patients in each year should be listed.

The mechanism of the significant result was not well illustrated in discussion.

Author Response

  1. A simple posthoc Analysis of Need for Second Procedures following Index Anterior and Posterior Urethroplasty. The finding is meaningful but not strong enough.

We thank the Reviewer for their feedback and the opportunity to improve our manuscript.

  1. The reality of this data is needed to be verified. Authors should upload the data as the supplement.

Data and analysis performed internally by TriNetX. As it is proprietary unable to upload the raw data. 

  1. Demographics and clinical characteristics of the study participants according to Anterior and Posterior Urethroplasty are missed.

We have added this information to the results section in paragraphs two and three.

“Patients were similar ages at index event (49.4 years for anterior urethroplasty and 49.3 years for anterior substitution urethroplasty). Most patients were White (66% and 76% for anterior and anterior substitution).”

“Patients with posterior urethroplasty were also of similar ages regardless of approach (49.1 years and 49.5 years for substitution urethroplasty). Again, most patients were White (72% and 82% for substitution).”

  1. CPTs are not needed to show in the table. x2 test should be performed in table 1.

We have removed the CPTs

x2 unable to be performed within TriNetX. TriNetX has capability for risk ratio as shown.

  1. KM figures listed a very wide 95% CI. Detailed numbers of patients in each year should be listed.

Numbers of patients at risk each year not available within TriNetX.

  1. The mechanism of the significant result was not well illustrated in discussion.

We have amended our discussion to further discuss our hypothesis. Please refer to the following addition to our limitations section

“Also, without access to the raw data we were unable to perform additional or separate analyses from those within TriNetX. The limitations of our study also restrict our ability to use the results to guide clinical practice. For example, we lack data on the size of stricture, both primary and recurrent, the severity of the stricture, as well as the size of graft or flap used and the indication for flap or graft. These factors are critical for preoperative planning as a surgeon decides which approach is correct for each patient. We thus caution interpretation of our comparisons between urethroplasty and substitution urethroplasty as our study lacks the stricture characteristics that are at least partially responsible for the outcomes presented.”

Round 2

Reviewer 2 Report

No more comments.